# From Text to Trajectory: Exploring Complex Constraint Representation and Decomposition in Safe Reinforcement Learning

**Pusen Dong[1], Tianchen Zhu[1], Yue Qiu[1], Haoyi Zhou[1,2], Jianxin Li[1,2]**
[1]Beihang University, Beijing, China
[2]Zhongguancun Laboratory, Beijing, China
`{dongps, zhutc, qiuyue, zhouhy, lijx}@act.buaa.edu.cn`

## Abstract

Safe reinforcement learning (RL) requires the agent to finish a given task while obeying specific constraints. Giving constraints in natural language form has great potential for practical scenarios due to its flexible transfer capability and accessibility. Previous safe RL methods with natural language constraints typically need to design cost functions manually for each constraint, which requires domain expertise and lacks flexibility. In this paper, we harness the dual role of text in this task, using it not only to provide constraint but also as a training signal. We introduce the Trajectory-level Textual Constraints Translator (TTCT) to replace the manually designed cost function. Our empirical results demonstrate that TTCT effectively comprehends textual constraint and trajectory, and the policies trained by TTCT can achieve a lower violation rate than the standard cost function. Extra studies are conducted to demonstrate that the TTCT has zero-shot transfer capability to adapt to constraint-shift environments.

## 1 Introduction

In recent years, reinforcement learning (RL) has achieved remarkable success in multiple domains, such as go game [37, 38] and robotic control [21, 45]. However, deploying RL in real-world scenarios still remains challenging. Many real-world decision-making applications, such as autonomous driving [13, 9] require agents to obey certain constraints while achieving the desired goals. To learn a safe constrained policy, some safe RL works [2, 33, 44, 49, 39, 48, 6] have proposed methods to maximize the reward while minimizing the constraint violations after training or during training.

However, several limitations prevent the existing safe RL methods' widespread use in real-world applications. Firstly, these methods often require mathematical or logical definitions of cost functions, which require domain expertise (**Limitation 1**). Secondly, their cost function definitions are frequently specific to a particular context and cannot be easily generalized to new tasks with similar constraints (**Limitation 2**). Lastly, most current safe RL methods focus on constraints that are logically simple, typically involving only one single entity or one single state [28], which can't represent the real-world safety requirements and lack universality (**Limitation 3**).

Using natural language to provide constraints [46, 29, 20] is a promising approach to overcome **Limitation 1** and **2** because natural language allows for flexible, high-level expression of constraints that can easily adapt to different scenarios. Regarding **Limitation 3**, previous approaches primarily employ what we call the *single state/entity textual constraint*. The single-state/entity textual constraint focuses solely on constraints related to one specific state or entity, limiting the ability to model complex safety requirements in real-world scenarios. Many safety requirements involve interactions and dependencies among multiple states or entities over time. By only addressing a single state or

Table 1: **Comparison of trajectory-level constraints and previous single state/entity constraints.** Universal Trajectory-level constraints can model any constraint requirements in real-world scenarios. But single state/entity constraint can only model the constraint requirements on individual state/entity.

| Single state/entity constraint (previous) | Trajectory-level constraint (ours) |
|---|---|
| Don't drive car. **(single entity)** | Don't drive car *after you drink wine*. **(multi entities)** |
| Don't touch lava. **(single state)** | Don't touch lava *more that three times*. **(multi states)** |
| Don't step in the gassed area. **(single state)** | Don't stay in the gas area for *more than 5 minutes*, your gas mask will fail. **(multi states)** |

entity, these constraints fail to capture the dynamic relationships and temporal aspects that are crucial for ensuring safety in complex environments. So we suggest using a more generalizable constraint type *trajectory-level textual constraint*. The trajectory-level textual constraint is a more universal constraint with complex logic semantics involving multiple states/entities. It is a nonlinear combination of multiple entities or multiple environment states and can model any constraint requirements in real-world scenarios. The trajectory-level constraints in natural language form are the highly abstract expression of the agent's behavior guidelines, serving as a more natural and straightforward way to introduce constraints. Notably, the set of trajectory-level constraints encompasses single-state constraints, as any single-state constraint can be reformulated as an equivalent trajectory-level constraint. Examples of trajectory-level textual constraint and single state/entity textual constraint are presented in Table 1.

Employing trajectory-level textual constraints across the entire trajectory poses two significant challenges. Firstly, determining whether an RL agent violates textual constraint over a trajectory is non-trivial, it needs to have the perception of the historical states and actions observed along the trajectory (**Challenge 1**). Secondly, the trajectory-level safety problem is susceptible to sparse cost [34], where cost is only imposed when the agent violates textual constraints at the final time step, making it challenging for the agent to learn which actions contribute to a gradual escalation of risk (**Challenge 2**). For instance, the agent needs to learn constraints like, "Don't touch lava after you touch grass." Without intermediate feedback, however, it struggles to understand how early actions, such as stepping on grass, contribute to eventual violations.

To address **Challenge 1**, we propose a general approach to align the trajectory's factual logic with the text's semantic logic, eliminating the need for manual encoding or separate models for each type of constraint. Our method employs a sequence model [41, 15] for modeling agent historical interactions with the environment, and a pre-trained language model (LM) [7] to comprehend natural language constraints. We then maximize the embedding similarities between matching pairs (trajectory, text) and minimize the embedding similarities between non-matching pairs using a contrastive learning approach [25] similar to CLIP [30]. By calculating the similarity between the textual embeddings of the constraint and the trajectory, we can predict whether a constraint is violated in this trajectory. Our method uniquely leverages text as both a source of constraints and a unified supervisory signal for trajectory encoding. In this dual role, text not only provides constraints but also guides the training process, enabling the model to naturally handle diverse semantic constraints without requiring specific model adjustments for each type. This design allows for a more flexible and generalizable system, significantly simplifying the handling of complex, multi-dimensional constraints.

In addition, to address the issue of cost sparsity (**Challenge 2**), we introduce a method for temporal credit assignment [40]. The proposed approach involves decomposing the one episodic cost of the textual constraint into multiple parts and allocating them to each state-action pair within the trajectory. This method offers denser cost signals regarding the relationship between the textual constraint and the agent's every action. It informs the agent which behaviors are risky and which are safer, thereby enhancing safety and aiding model performance.

Our experiments demonstrate that the proposed method can effectively address **Challenge 1** and **Challenge 2**. In both 3D navigation [19] and 2D grid exploration [46] tasks, agents trained using our method achieve significantly lower violation rates (up to 4.0x) compared to agents trained with ground-truth cost functions while maintaining comparable rewards and more importantly, our method obtains the Pareto frontier [10]. In addition to this, our method has zero-shot adaptation capability to adapt to constraint-shift environments without fine-tuning.

## 2   Related Work

**Safe RL.** Safe RL aims to train policies that maximize reward while minimizing constraint violations [11, 13]. In prior work, there are usually two ways to learn safe policies: (1) consider cost as one of the optimization objectives to achieve safety [48, 6, 49, 39, 44, 33], and (2) achieve safety by leveraging external knowledge (e.g. expert demonstration) [35, 32, 1, 47]. These typical safe RL algorithms require either human-defined cost functions or human-specified cost constraints which are unavailable in the tasks that constraints are given by natural language.

**RL with Natural Language.** Prior works have integrated natural language into RL to improve generalization or learning efficiency in various ways. For example, Hermann et al. [14] studied how to train an agent that can follow natural language instructions to reach a specific goal. Additionally, natural language has been used to constrain agents to behave safely. For instance, Prakash et al. [29] trained a constraint checker to predict whether natural language constraints are violated. Yang et al. [46] trained a constraint interpreter to predict which entities in the environment may be relevant to the constraint and used the interpreter to predict costs. Lou et al. [28] used pre-trained language models to predict the cost of specific states, avoiding the need for artificially designed cost functions. However, previous methods cannot uniformly handle textual constraints with one framework, which limits their applicability.

**Credit Assignment in RL.** Credit assignment studies the problem of inferring the true reward from the designed reward. Prior works have studied improving sample efficiency of RL algorithms through credit assignment, for example by using information gain [17], as an intrinsic bonus reward to aid exploration. Goyal et al. [12] proposed the use of natural language instructions to perform reward shaping to improve the sample efficiency of RL algorithms. Liu et al. [26] learned to decompose the episodic return as the reward for policy optimization. However, to the best of our knowledge, our work is the first to apply credit assignment to safe RL.

## 3   Preliminaries

**Problem formulation.** Trajectory-level constraint problem can be formed as the Constrained Non-Markov Decision Process (CNMDP) [3, 43], and it can be defined by the tuple $<S, A, T, R, \gamma, C, Y, \tau^*>$. Here $S$ represents the set of states, $A$ represents the set of actions, $T$ represents the state transition function, $R$ represents the reward function, and $\gamma \in (0, 1)$ represents the discount factor. In addition, $Y$ represents the set of trajectory-level textual constraints (e.g., "You have 10 HP, you will lose 3 HP every time you touch the lava, don't die."), which describes the constraint that the agent needs to obey across the entire trajectory. $C$ represents the cost function determined by $y \in Y$. $\tau^*$ represents the set of historical trajectories.

**RL with constraints.** The objective for the agent is to maximize reward while obeying the specified textual constraint as much as possible [27]. Thus, in our task setting, the agent needs to learn a policy $\pi\colon S \times Y \times \tau^* \to P(A)$ which maps from the state space $S$, textual constraints $Y$ and historical trajectories $\tau^*$ to the distributions over actions $A$. Given a $y$, we learn a policy $\pi$ that maximizes the cumulative discounted reward $J_R$ while keeping the cumulative discounted cost (average violation rate) $J_C$ below a constraint violation budget $B_C(y)$:

$$\max_\pi J_R(\pi) = \mathbb{E}_\pi\big[\sum_{t=0}^\infty \gamma R(s_t)\big] \quad \text{s.t.} \quad J_C(\pi) = \mathbb{E}_\pi\big[\sum_{t=0}^\infty \gamma C(s_t, a_t, y, \tau_t)\big] \leq B_C(y). \tag{1}$$

Here $B_C(y)$ and $C(s_t, a_t, y, \tau_t)$ are two functions both depending on textual constraint $y$. $\tau_t$ represents the historical trajectory at time step $t$.

**Episodic RL.** Similar to the task with episodic rewards [4], in our task setting, a cost is only given at the end of each trajectory $\tau$ when the agent violates the textual constraint $y$. In other words, before violating $y$, the cost $C(s_t, a_t, y, \tau_t) = 0$ for all $t < T$. For simplicity, we omit the discount factor and assume that the trajectory length is at most $T$ so that we can denote $a_T$ as the final action that causes the agent to violate $y$ without further confusion. Therefore, the constraint qualification of the objective in RL with constraints becomes $J_C(\pi) = \mathbb{E}_\pi[C(s_T, a_T, y, \tau_T)] \leq B_C(y)$. Due to the sparsity of cost, a large amount of rollout trajectories are needed to help the agent distinguish the subtle effects of actions on textual constraint [24]. This situation will become more serious when trajectory-level constraints are complex and difficult to understand.

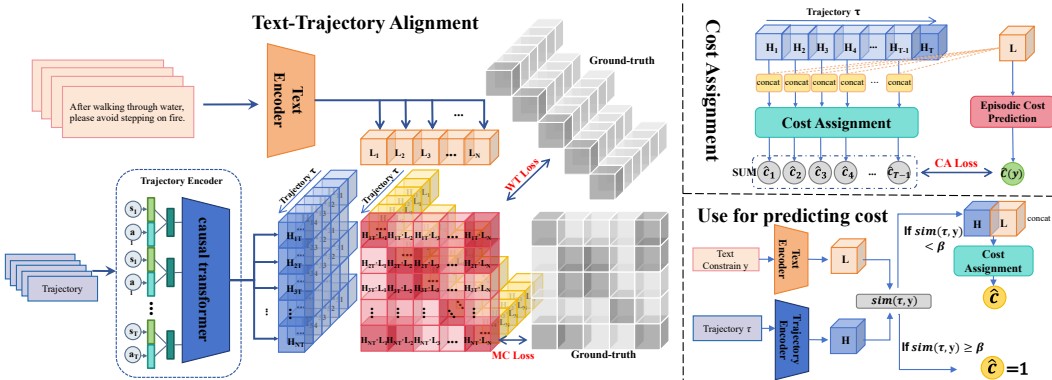

Figure 1: **TTCT overview.** TTCT consists of two training components: (1) *the text-trajectory alignment component* connects trajectory to text with multimodal architecture, and (2) *the cost assignment component* assigns a cost value to each state-action based on its impact on satisfying the constraint. When training RL policy, the text-trajectory alignment component is used to predict whether a trajectory violates a given constraint and the cost assignment component is used to assign non-violation cost.

## 4    TTCT: Trajectory-level Textual Constraints Translator

In this section, we introduce our proposed framework **TTCT** ( **T**rajectory-level **T**extual **C**onstraints **T**ranslator) as shown in Figure 1. TTCT has two key components: *the text-trajectory alignment component* and *the cost assignment component*. The text-trajectory alignment component is used to address the violations prediction problem. The cost assignment component is used to address the sparse cost problem.

### 4.1    Text-Trajectory Alignment Component

We propose a component to learn from offline data to predict whether a given trajectory violates textual constraints. The core idea of this component is to learn trajectory representations under textual supervision and connect trajectory representation to text representation. If the distance between the two representations in the embedding space is sufficiently close, we can consider that the given trajectory violates the constraint. Our approach does not require modeling entities of the environment like previous work, such as [46], which involves labeling hazardous items artificially in every observation. Instead, we model this task as a trajectory-text multimodal learning problem. Hence, our method can learn trajectory representations and text representations from the pairs *(trajectory, trajectory-level textual constraint)*. We believe that learning from the supervision of natural language could not only enhance the representation power but also enable flexible zero-shot transfer [30].

Formally, given a batch of $N$ (trajectory $\tau$, trajectory-level textual constraint $y$) pairs. For each pair, the trajectory corresponds to the text, indicating that the given trajectory violates the given textual constraint. The trajectory can be defined as $\tau = (s_1, a_1, s_2, a_2, ..., s_{T-1}, a_{T-1}, s_T, a_T)$, where $T$ is the step at which the textual constraint $y$ is first violated by the trajectory. Here $s_t$ is a $d_s$-dimensional observation vector. Each state in the trajectory is processed by a state encoder to obtain a representation $v_t^s$, also action is processed by an action encoder to obtain a representation $v_t^a$. Then, we concatenate $v_t^s$ and $v_t^a$ to obtain a vector representation $v_t$ for each state-action pair. After that, we learn separate unimodal encoders $g_T$ and $g_C$ for the trajectory and textual constraint, respectively. The trajectory encoder $g_T$ utilizes a causal transformer to extract the trajectory representation from the input state-action representation sequence $\{v_t\}_{t=1}^T$:

$$H_1, H_2, H_3, ..., H_{T-1}, H_T = g_T(\{v_t\}_{t=1}^T),  \tag{2}$$

where $H_t$ is a $d_H$-dimensional vector. The final embedding $H_T$ is used as the representation for the entire trajectory. Specifically, the causal Transformer processes the trajectory sequence by maintaining a left-to-right context while generating embeddings. This allows the model to capture temporal dependencies within the trajectory and obtain the embeddings for time steps before $T$. The textual constraint encoder $g_C$ is used to extract features that are related to the constraints and it could be one of a wide variety of language models:

$$L = g_C(y),  \tag{3}$$

where $L$ is a $d_L$-dimensional vector representation. Then we define symmetric similarities between the two modalities with cosine distance:

$$\text{sim}_T(\tau, y) = \exp(\alpha) * \frac{H_T \cdot L^\top}{\|H_T\| \, \|L\|}, \quad \text{sim}_T(y, \tau) = \exp(\alpha) * \frac{L^\top \cdot H_T}{\|L\| \, \|H_T\|}, \tag{4}$$

where $\alpha$ is a learnable parameter and $T$ means that use last embedding of trajectory to calculate similarity. And we use softmax to calculate $trajectory \to text$ and $text \to trajectory$ similarities scores:

$$p_i^{\tau \to y}(\tau) = \frac{\exp(\text{sim}_T(\tau, y_i))}{\sum_{j=1}^{N} \exp(\text{sim}_T(\tau, y_j))}, \quad p_i^{y \to \tau}(y) = \frac{\exp(\text{sim}_T(y, \tau_i))}{\sum_{j=1}^{N} \exp(\text{sim}_T(y, \tau_j))}. \tag{5}$$

Let $q^{\tau \to y}(\tau), q^{y \to \tau}(y)$ indicate the ground-truth similarity scores, where the **negative pair** (trajectory doesn't violate textual constraint) has a probability of 0 and the **positive pair** (trajectory violate textual constraint) has a probability of 1. In our task setting, a trajectory can correspond to multiple textual constraints, and vice versa. For example, two textual constraints such as *"Do not touch lava"* and *"After stepping on water, do not touch lava"* might both be violated by a single given trajectory. This many-to-many relationship between trajectories and textual constraints implies that the same textual constraints (or different textual constraints with the same semantics) can apply to multiple trajectories, while a single trajectory may comply with several textual constraints. So there may be more than one positive pair in $q_i^{\tau \to y}(\tau)$ and $q_i^{y \to \tau}(y)$. Therefore, we use Kullback–Leibler (KL) divergence [23] as the **multimodal contrastive (MC)** loss to optimize our encoder similar to [42]:

$$\mathcal{L}_{MC} = \frac{1}{2} \mathbb{E}_{(\tau, y) \sim \mathcal{D}} [\text{KL}(p^{\tau \to y}(\tau), \text{softmax}(q^{\tau \to y}(\tau))) + \text{KL}(p^{y \to \tau}(y), \text{softmax}(q^{y \to \tau}(y)))], \tag{6}$$

where $\mathcal{D}$ is the training set.

In addition to the multimodal contrastive (MC) loss, we also introduce a **within-trajectory (WT)** loss. Specifically, suppose we have a trajectory's representation sequence $(H_1, H_2, H_3, ..., H_{(T-1)}, H_T)$ and its corresponding textual constraint embedding $L$, we can calculate cosine similarity $\text{sim}_t(\tau, y)$ between embedding $H_t$ and $L$ using Equation 4. Then we can calculate similarity scores within the trajectory:

$$p_t^\tau(y) = \frac{\exp(\text{sim}_t(\tau, y))}{\sum_{k=1}^{T} \exp(\text{sim}_k(\tau, y))}. \tag{7}$$

Different from $p_i^{y \to \tau}(y)$ in Equation 5, which measures similarity scores across $N$ trajectories, Equation 7 is used to measure the similarity scores of different time steps within a trajectory. The reason for doing this is that the textual constraint is violated at time step $T$, while in the previous time steps the constraint is not violated. Therefore, the instinct is to maximize the similarity score between the final trajectory embedding $H_T$ and the textual constraint embedding $L$, while minimizing the similarity score between all previous time step embeddings $(H_1, H_2, H_3, ..., H_{(T-1)})$ and the textual constraint embedding $L$. Based on this instinct, we introduce **within-trajectory (WT)** loss:

$$\mathcal{L}_{WT} = \mathbb{E}_{(\tau, y) \sim \mathcal{D}} \left[ -\frac{1}{T} \left( \sum_{t=1}^{T-1} \log(1 - p_t^\tau(y)) + \log(p_T^\tau(y)) \right) \right], \tag{8}$$

where the first term is responsible for minimizing the similarity score for the embedding of time steps before $T$ in the trajectory sequence, while the second term is responsible for maximizing the similarity score for the embedding of time step $T$.

By combining these two losses $\mathcal{L}_{WT}, \mathcal{L}_{MC}$, we can train a text encoder to minimize the distance between embeddings of semantically similar texts, while simultaneously training a trajectory encoder to minimize the distance between embeddings of semantically similar trajectories. Crucially, this approach enables us to align the text and trajectory embeddings that correspond to the same semantic constraint, fostering a cohesive representation of their shared meaning, and further determining whether the trajectory violates the constraint by calculating embedding similarity.

## 4.2 Cost Assignment Component

After solving the problem of predicting whether a given trajectory violates a textual constraint, there is still the issue of cost sparsity. Motivated by the works of temporal credit assignment [26], we propose

a component to capture the relationship between the state-action pair and the textual constraint and assign a cost value to each state-action based on its impact on satisfying the constraint.

Specifically, suppose we have a (trajectory $\tau$, textual constraint $y$) pair and its representation $(\{H_t\}_{t=1}^{T}, L)$ obtained from text-trajectory alignment component. The textual constraint representation $L$ is processed by an episodic-cost prediction layer $F^e$ to obtain a predicted episodic cost $\hat{C}(y) =$ sigmoid$(F^e(L))$ for the entire trajectory. We expect the episodic cost can be considered as the sum of cost on all non-violation state-action pairs: $\hat{C}(y) = \sum_{t=1}^{T-1} \hat{c}(s_t, a_t, y, \tau_t)$. To evaluate the significance of each timestep's action relative to textual constraints, we employ an attention mechanism:

$$e_t = \text{sigmoid}(\text{sim}_t(\tau, y)). \tag{9}$$

Here we regard the text representation as the $query$, each time step's representation in the trajectory as the $key$, and compute the attention score $e_t$ based on the cosine similarity metric. After that, we use the sigmoid function to make sure the score falls within the range of 0 to 1. Each attention score $e_t$ quantifies the degree of influence of the state-action pair $(s_t, a_t)$ on violating the textual constraint. Then we obtain an influence-based representation $H_t^* = e_t H_t$. To predict the cost $\hat{c}(s_t, a_t, y, \tau_t)$, we incorporate a feed-forward layer called the cost assignment layer $F^c$ and output the predicted non-violation single step cost as:

$$\hat{c}(s_t, a_t, y, \tau_t) = \text{sigmoid}(F^c(\text{Concat}(H_t^*, L))). \tag{10}$$

The loss function for measuring inconsistency of the episodic cost $\hat{C}(y)$ and the sum of non-violation single step costs $\hat{c}(s_t, a_t, y, h_t)$ can be formed as:

$$\mathcal{L}_{CA} = \mathbb{E}_{(\tau, y) \sim \mathcal{D}}\left[\left(\sum_{t=1}^{T-1} \hat{c}(s_t, a_t, y, \tau_t) - \hat{C}(y)\right)^2\right]. \tag{11}$$

This mutual prediction loss function $\mathcal{L}_{CA}$ is only used to update the episodic-cost prediction layer and cost assignment layer, and not to update the parameters of the trajectory encoder or text encoder during backpropagation. This helps ensure the validity of the predictions by preventing overfitting or interference from other parts of the model.

The effectiveness of this component comes from two main sources. First, the text-trajectory alignment component projects semantically similar text representations to nearby points in the embedding space, allowing the episodic-cost prediction layer to assign similar values to embeddings with close distances. This aligns with the intuition that textual constraints with comparable violation difficulty should yield similar episodic costs. Second, the cost assignment layer leverages the representational power of the text-trajectory alignment component to capture complex relationships between state-action pairs and constraints, enabling accurate single-step cost predictions.

In our experiment, we simultaneously train the text-trajectory alignment component and cost assignment component end-to-end with a uniform loss function $\mathcal{L}_{TTCT}$:

$$\mathcal{L}_{TTCT} = \mathcal{L}_{MC} + \mathcal{L}_{WT} + \mathcal{L}_{CA}. \tag{12}$$

By doing this, we can avoid the need for separate pre-training or fine-tuning steps, which can be time-consuming and require additional hyperparameter tuning. Also, this can enable the cost assignment component to gradually learn from the text-trajectory alignment component and make more accurate predictions over time.

In the test phase, at time step $t$ we encode trajectory $\tau_t$ and textual constraint $y$ with Equation 3 and Equation 2 to obtain the entire trajectory embedding $H_t$ and text embedding $L$. Then we calculate distance score $\text{sim}(\tau_t, y)$ using Equation 4. The predicted cost function $\hat{c}$ is given by:

$$\hat{c} = \begin{cases} 1, & \text{if } \text{sim}(\tau_t, y) \geq \beta, \\ \text{sigmoid}(F^c(\text{Concat}(H_t^*, L))), & \text{otherwise} \end{cases}, \tag{13}$$

where $\beta$ is a hyperparameter that defines the threshold of the cost prediction. $\hat{c} = 1$ indicates that the TTCT model predicts that the textual constraint is violated. Otherwise, if the textual constraint is not violated by the given trajectory, we assign a predicted cost using Equation 10.

# 5   Policy Training

Our Trajectory-level textual constraints Translator framework is a general method for integrating free-form natural language into safe RL algorithms. In this section, we introduce how to integrate our TTCT into safe RL algorithms so that the agents can maximize rewards while avoiding early termination of the environment due to violation of textual constraints. To enable perception of historical trajectory, the trajectory encoder and text encoder are not only used as **frozen plugins** $g_T$ and $g_C$ for cost prediction but also as **trainable sequence models** $g_T^*$ and $g_C^*$ for modeling historical trajectory. This allows the agent to take into account historical context when making decisions. To further improve the ability to capture relevant information from the environment, we use LoRA [18] to fine-tune both the $g_T^*$ and $g_C^*$ during policy training. The usage of $g_T$, $g_C$ and $g_T^*$, $g_C^*$ is illustrated in Appendix A.4 Figure 8.

Formally, let's assume we have a policy $\pi$ with parameter $\phi$ to gather transitions from environments. We maintain a vector to record the history state-action pairs sequence, and at time step $t$ we use $g_T^*$ and $g_C^*$ to encode $\tau_{t-1}$ and textual constraint $y$ so that we can get historical context representation $H_{t-1}$ and textual constraint representation $L$. The policy selects an action $a_t = \pi_\theta(o_t, H_{t-1}, L)$ to interact with environment to get a new observation $o_{t+1}$. And we update $\tau_t$ with the new state-action pair $(o_t, a_t)$ to get $\tau_t$. With $\tau_t$ and $L$, $\hat{c}_t$ can be predicted according to Equation 13. Then we store the transition into the buffer and keep interacting until the buffer is full. In the policy updating phase, after calculating the specific loss function for different safe RL algorithms, we update the policy $\pi$ with gradient descent and update $g_T^*$, $g_C^*$ with LoRA. It is worth noting that $g_T$ and $g_C$ are not updated during the whole policy training phase, as they are only used for cost prediction. The pseudo-code and more details of the policy training can be found in Appendix A.4.

# 6   Experiments

Our experiments aim to answer the following questions: **(1)** Can our TTCT accurately recognize whether an agent violates the trajectory-level textual constraints? **(2)** Does the policy network, trained with predicted cost from TTCT, achieve fewer constraint violations than trained with the ground-truth cost function? **(3)** How much performance improvement can the cost assignment (CA) component achieve? **(4)** Does our TTCT have zero-shot capability to be directly applicable to constraint-shift environments without any fine-tuning? We adopt the following experiment setting to address these questions.

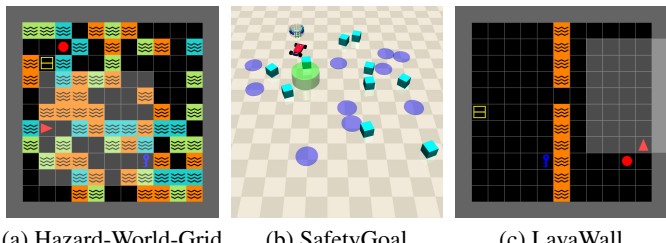

(a) Hazard-World-Grid     (b) SafetyGoal     (c) LavaWall

Figure 2: (a) One layout in **Hazard-World-Grid** [46], where orange tiles are lava, blue tiles are water and green tiles are grass. Agents need to collect reward objects in the grid while avoiding violating our designed textual constraint for the entire episode. (b) Robot navigation task **SafetyGoal** that is built in Safety-Gymnasium [19], where there are multiple types of objects in the environment. Agents need to reach the goal while avoiding violating our designed textual constraint for the entire episode. (c) **LavaWall** [5], a task has the same goal but different hazard objects compared to Hazard-World-Grid.

## 6.1   Setup

**Task.** We evaluate TTCT on two tasks (Figure 2 (a,b)): 2D grid exploration task *Hazard-World-Grid* (Grid) [46] and 3D robot navigation task *SafetyGoal* (Goal) [19]. And we designed over 200 trajectory-level textual constraints which can be grouped into 4 categories, to constrain the agents. A detailed description of the categories of constraints will be given in Appendix A.1. Different from the default setting, in our task setting, when a trajectory-level textual constraint is violated, the

environment is immediately terminated. This is a more difficult setup than the default. In this setup, the agents must collect as many rewards as possible while staying alive.

**Baselines.** We consider the following baselines: `PPO` [36], `PPO_Lagrangian(PPO_Lag)` [33], `CPPO_PID` [39], `FOCOPS` [49]. `PPO` does not consider constraints and simply aims to maximize the average reward. We use `PPO` to compare the ability of our methods to obtain rewards. As for the last three algorithms, we design two training modes for them. One is trained with standard ground-truth cost, where the cost is given by the human-designed violation checking functions, and we call it ground-truth (GC) mode. The other is trained with the predicted cost by our proposed TTCT, which we refer to as cost prediction (CP) mode. More information about the baselines and training modes can be found in Appendix A.1 A.2.

**Metrics.** We take average episodic reward (Avg. R) and average episodic cost (Avg. C) as the main comparison metrics. Average episodic cost can also be considered as the average probability of violating the constraints. The higher Avg. R, the better performance, and the lower Avg. C, the better performance.

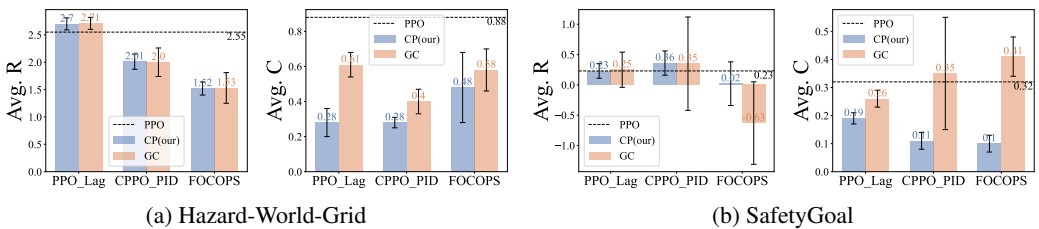

|     |     |
| :-: | :-: |
| (a) Hazard-World-Grid | (b) SafetyGoal |

Figure 3: **Evaluation results of our proposed method TTCT.** The blue bars are our proposed cost prediction (CP) mode performance and the orange bars are the ground-truth cost (GC) mode performance. The black dashed lines are `PPO` performance. (a) Results on Hazard-World-Grid task. (b) Results on SafetyGoal task.

## 6.2 Main Results and Analysis

The evaluation results are shown in Figure 3 and the learning curves are shown in Figure 4. We can observe that in the *Hazard-World-Grid* task, compared with `PPO`, the policies trained with GC can reduce the probability of violating textual constraints to some extent, but not significantly. This is because the sparsity of the cost makes it difficult for an agent to learn the relevance of the behavior to the textual constraints, further making it difficult to find risk-avoiding paths of action. In the more difficult *SafetyGoal* task, it is even more challenging for GC-trained agents to learn how to avoid violations. In the `CPPO_PID` and `FOCOPS` algorithms trained with GC mode, the probability of violations even rises gradually as the training progresses. In contrast, the agents trained with predicted cost can achieve lower violation probabilities than GC-trained agents across all algorithms and tasks and get rewards close to GC-trained agents.

These results show that **TTCT can give an accurate predicted episodic cost at the time step when the constraint is violated, it can also give timely cost feedback to non-violation actions through the cost assignment component so that the agents can find more risk-averse action paths.** And these results answer questions **(1)** and **(2)**. The discussion about the violation prediction capability of the text-trajectory component can be found in Appendix B.1. The interpretability and case study of the cost assignment component can be found in Appendix B.2.

## 6.3 Ablation Study

To study the influence of the cost assignment component. We conduct an ablation study by removing the cost assignment component from the full TTCT. The results of the ablation study experiment are shown in Figure 4. We can observe that even TTCT without cost assignment can achieve similar performance as GC mode. And in most of the results if we remove the cost assignment component, the performance drops. This shows that **our text trajectory alignment component can accurately predict the ground truth cost, and the use of the cost assignment component can further help us learn a safer agent**. These results answer questions **(3)**.

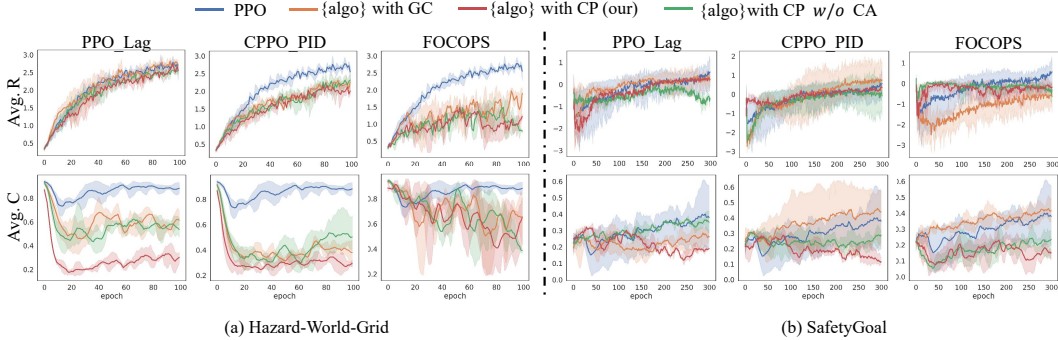

Figure 4: **Learning curve of our proposed method TTCT.** Each column is an algorithm. The six figures on the left show the results of experiments on the Hazard-World-Grid task and the six figures on the right show the results of experiments on the SafetyGoal task. The solid line is the mean value, and the light shade represents the area within one standard deviation.

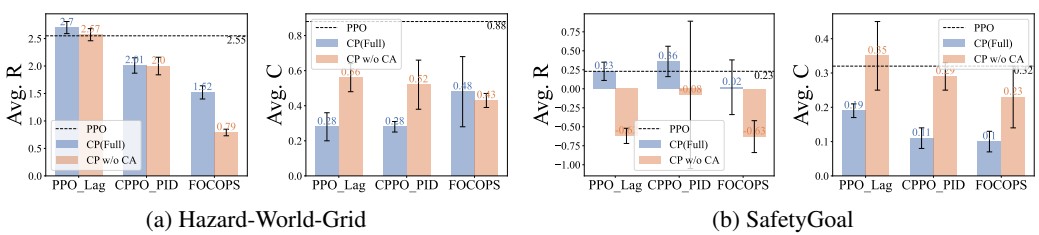

Figure 5: **Ablation study of removing the cost assignment (CA) component.** The blue bars are cost prediction (CP) mode performance with full TTCT and the orange bars is the cost prediction (CP) mode performance without CA component. The black dashed lines are PPO performance. (a) Ablation results on Hazard-World-Grid task. (b) Ablation results in SafetyGoal task.

## 6.4 Further Results

**Pareto frontier.** Multi-objective optimization typically involves finding the best trade-offs between multiple objectives. In this context, it is important to evaluate the performance of different methods based on their Pareto frontier [10], which represents the set of optimal trade-offs between the reward and cost objectives. We plot the Pareto frontier of policies trained with GC and policies trained with CP on a two-dimensional graph, with the vertical axis representing the reward objective and the horizontal axis representing the cost objective as presented in Figure 6. The solution that has the Pareto frontier closer to the origin is generally considered more effective than those that have the Pareto frontier farther from the origin. We can observe from the figure that the policies trained with predicted cost by our TTCT have a Pareto frontier closer to the origin. This proves the effectiveness of our method and further answers Questions **(1)** and **(2)**.

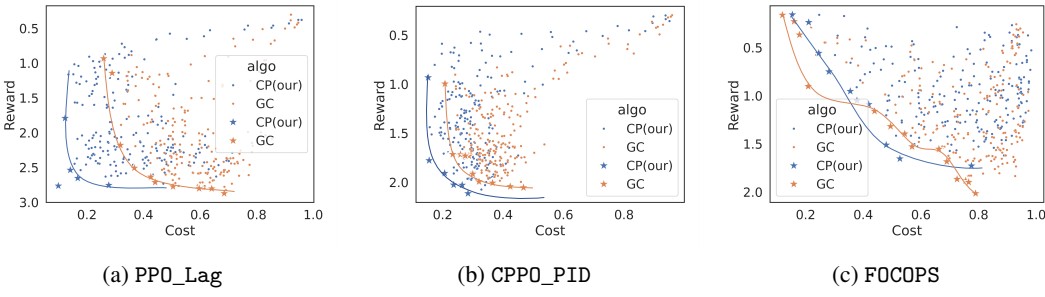

Figure 6: **Results of Pareto frontiers.** We compare the performance of 200 policies trained using cost prediction (CP) and 200 policies trained with ground-truth cost (GC). The ★ symbol represents the policy on the Pareto frontier. And we connect the Pareto-optimal policies with a curve.

**Zero-shot transfer capability.** To explore whether our method has zero-shot transfer capability, we use the TTCT trained under the *Hazard-World-Grid* environment to apply directly to a new environment called *LavaWall* (Figure 2 (c)) [5], without fine-tuning. The results are shown in Figure 7. We can observe that the policy trained with cost prediction (CP) from TTCT trained under the Hazard-World-Grid environment can still achieve a low violation rate comparable to the GC-trained policy. This answers Question **(4)**.

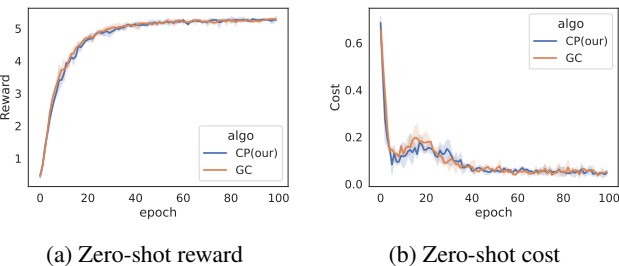

(a) Zero-shot reward      (b) Zero-shot cost

Figure 7: **Zero-shot adaptation capability of TTCT on LavaWall task.** The left figure shows the average reward and the right figure shows the average cost.

## 7 Conclusion and Future Work

In this paper, we study the problem of safe RL with trajectory-level natural language constraints and propose a method of trajectory-level textual constraints translator (TTCT) to translate constraints into a cost function. By combining the text-trajectory alignment (CA) component and the cost assignment (CA) component, our method can elegantly solve the problems of predicting constraint violations and cost sparsity. We demonstrated that our TTCT method achieves a lower violation probability compared to the standard cost function. Thanks to its powerful multimodal representation capabilities, our method also has zero-shot transfer capability to help the agent safely explore the constraint-shift environment. This work opens up new possibilities for training agents in safe RL tasks with total free-form and complex textual constraints.

Our work still has room for improvement. The violation rate of our method is not absolute zero. In future work, we plan to investigate the application of TTCT in more complex environments and explore the integration of other techniques such as meta-learning [16] to further improve the performance and generalization capabilities of our method.

## Acknowledgments

This work was supported by the grants from the Natural Science Foundation of China (62225202, 62202029), and Young Elite Scientists Sponsorship Program by CAST (No. 2023QNRC001). Thanks for the computing infrastructure provided by Beijing Advanced Innovation Center for Big Data and Brain Computing. This work was also sponsored by CAAI-Huawei MindSpore Open Fund. Jianxin Li is the corresponding author.

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

# A  Dataset and Training Details

## A.1  Dataset

In our task setting, humans need to provide high-level textual instruction to the agent for the entire trajectory, and then TTCT can predict the cost based on the real-time state of the agent's exploration so that the agent can learn a safe policy with the predicted cost. Thus our dataset is comprised of two parts: the trajectory-level textual constraints and the environments.

**Trajectory-level textual constraints:** To generate textual constraints, we first explore the environment using a random policy, collecting a large amount of offline trajectory data. Then we design a descriptor that automatically analyzes trajectories and gives natural language descriptions based on predefined templates. To validate whether our method can understand different difficulty levels of textual constraints, we design four types of trajectory-level textual constraints. The four types of textual constraints are:

1. **Quantitative** textual constraint describes a quantitative relationship in which an entity in the environment cannot be touched beyond a specific number of times, which can be interpreted as the entity's tolerance threshold, and when the threshold is exceeded, the entity may experience irrecoverable damage.

2. **Sequential** textual constraint describes a sequence-based relationship, where the occurrence of two or more distinct actions independently may not pose a risk, but when they occur in sequence, it does. For instance, it's safe to drink or drive when they happen independently, but when they occur in sequence (i.e., drinking first and then driving), it becomes dangerous.

3. **Relational** textual constraint describes constraints on the relationships between an agent and entities in its environment, such as maintaining a certain distance, always being in front of that entity, or not staying too far from it.

4. **Mathematical** textual constraints often do not provide explicit instructions to the agent regarding what actions to avoid, but rather present logical descriptions that demand the model's ability to reason mathematically. This type of constraint thereby presents a higher cognitive burden for our TTCT to comprehend.

Examples of four types of trajectory-level textual constraints are included in Table 2. Then we randomly split the (trajectory, textual constraint) pairs into $80\%$ training and $20\%$ test sets. And we use the training set to train our TTCT end-to-end.

**Environments:** We use two environments **Hazard-World-Grid** and **SafetyGoal** as main benchmarks and a environment **LavaWall** to evaluate the zero-shot transfer capability:

1. **Hazard-World-Grid.** The environment is a $12 \times 12$ grid, with the gray walls surrounding the perimeter. The agent can only explore within the grid. Inside the grid, some items provide rewards: blue keys, red balls, and yellow boxes. Collecting all of these items will be considered as completing the task. Additionally, there are hazardous materials in the grid, where orange tiles are lava, cyan tiles are water, and green tiles are grass. During exploration, the agent can only see a range of $7 \times 7$ pixels ahead, resulting in an observation space with size $7 \times 7 \times 3$.

2. **SafetyGoal.** Robot navigation task, the environment has a navigation target, which is completed when the robot reaches the target. The environment also contains vases (cyan squares) and Hazards (Mediumslateblue circles). The vases can be moved by the robot and hazards are fixed.

3. **LavaWall.** The LavaWall environment shares the same task goal as Hazard-World-Grid but with different hazardous materials. Whenever the environment is initialized, a random lava wall with only one entrance is generated in the grid, and the agent must learn to obtain the reward on the other side while avoiding stepping on lava.

For each episode, we place the agent at a randomized start location, fill the environment with objects, and randomly select a textual constraint from the constraint pool. We manually design the violation checking function for every constraint to determine whether the environment violates the textual constraint, and when the textual constraint is violated, the environment will provide a cost with a

Table 2: **Examples from the different trajectory-level textual constraint categories.**

| Constraint type | Examples |
|---|---|
| **Quantitative** | Do not cross lava more than five times. |
| | Never reach grass more than five times. |
| | Please touch water less than three times. |
| | The vase is very fragile, please don't touch it! |
| | If you touch the vase more than three times, the vase will break. |
| | The lava is very dangerous. Please don't touch it. |
| **Sequential** | After stepping through the water, your shoes can't touch the lava. |
| | Once you've walked past the grass, make sure you don't set foot on water. |
| | After walking past lava, don't go back to the water. |
| **Mathematical** | You only have 20 HP, grass and lava are very dangerous, you will lose 3 and 2 HP respectively if you step on them, please don't die! |
| | You only have 25 HP, stepping on grass and lava will lose 3 and 2 HP respectively, but water will help you regain 1 HP, please don't die! |
| | You only have 10 HP, and stepping on dangerous water and lava will cause you to lose 3 and 2 HP respectively, your fate hangs by a thread. don't let it snap! |
| **Relational** | Please keep a distance of 0.2 from dangerous chemicals or you will die! |
| | Maintain a safe distance of 0.25 from the hazard. |
| | It's important to keep a distance of 0.3 from the hazard. |

value of 1.0 to the agent and terminate the environment (the manually designed function will not be used in the cost prediction (CP) mode, all the costs will be predicted by our TTCT in CP mode).

## A.2 Baselines

(1) PPO [36]: This algorithm does not consider constraints, and simply considers maximizing the average reward, which we use to compare the ability of our methods to obtain rewards.
(2) PPO_Lagrangian(PPO_Lag) [33]: This algorithm transforms a constrained optimization problem into an unconstrained optimization problem via Lagrange multipliers
(3) CPPO_PID [39]: This algorithm PID to control Lagrange multiplier, which solves the cycle fluctuation problem of PPO-Lagrange.
(4) FOCOPS [49]: This algorithm finds the optimal update policy by solving a constrained optimization problem in the nonparameterized policy space, then projects the update policy back into the parametric policy space.

## A.3 Training Detail of TTCT

We provide training pseudocode of TTCT in Figure 9. The text encoder we use is Bert ((https://huggingface.co/google-bert/bert-base-uncased/tree/main) [7]. And the hyperparameters we use are shown in Table 3. We use the same hyperparameters across different tasks' datasets. We conduct the training on the machine with A100 GPU for about 1 day.

## A.4 Policy Training Detail

We provide pseudocode of training policy with predicted from TTCT in Algorithm 1. To enable the agent to comply with diverse types of constraints, we launch async vectorized environments with different types of constrained environments during roll-out collection. The agent interacts with these environments and collects transitions under different constraint restrictions to update its policy. During policy updates, we fine-tune the trajectory encoder $g_H^*$ and text encoder $g_L^*$ using LoRA [18] at every epoch's first iteration, while not updating encoder parameters in other iterations, which saves training time.

Table 3: Hyperparameters used in TTCT

| Hyperparameters | |
|---|---|
| Batch size | 194 |
| Epochs | 32 |
| Learning rate | $10^{-6}$ |
| Embedding dimension of trajectory $d_H$ | 512 |
| Embedding dimension of text $d_T$ | 512 |
| Trajectory length | 200 |
| Transformer width | 512 |
| Transformer number of heads | 8 |
| Transformer number of layers | 12 |
| Optimizer | Adam [22] |

---

**Algorithm 1** Safe RL algorithm with TTCT

---

1: Initialize value function network $\phi$, cost value function network $\phi_c$, policy network $\theta$, trajectory encoder $g_T^*$, textual constraint encoder $g_C^*$, frozen trajectory encoder $g_T$ for cost prediction, frozen textual constraint encoder $g_C$ for cost prediction, cost alignment network $F^c$
2: **for** each epoch **do**
3:     **for** each episode **do**
4:         Sample a textual constraint $y$ for this episode
5:         Get text representation $L$ of $y$ with $g_C^*$
6:         Reset environment to get observation $o_1$
7:         Initialize sequence $\tau_0 = \{o_{pad}, a_{pad}\}$ and encoded sequence representation $H_0 = g_T^*(\tau_1)$
8:         **for** $t = 1, T$ **do**
9:             Select a action $a_t = \pi_\theta(o_t, H_{t-1}, L)$
10:             Execute action $a_t$ in emulator and observe reward $r_t$ and next observation $o_{t+1}$
11:             Append $(o_t, a_t)$ into $\tau_{t-1}$ to get $\tau_t$ and update sequence representation $H_t = g_T^*(\tau_t)$
12:             Predict cost $\hat{c}_t$ according to Equation 13
13:             Store transition $(o_t, a_t, r_t, \hat{c}_t, \tau_{t-1}, y)$ in buffer
14:         **end for**
15:     **end for**
16:     Sample batch of $N$ transitions from buffer.
17:     Encode $\{\tau_j\}_{i=1}^N$ and $\{y_i\}_{i=1}^N$ with $g_T, g_C$
18:     Calculate specific loss function
19:     Update value function $\phi$, cost value function $\phi_c$, policy network $\theta$
20:     Update trajectory encoder $g_T^*$ and textual constraint encoder $g_C^*$ with LoRA
21: **end for**

---

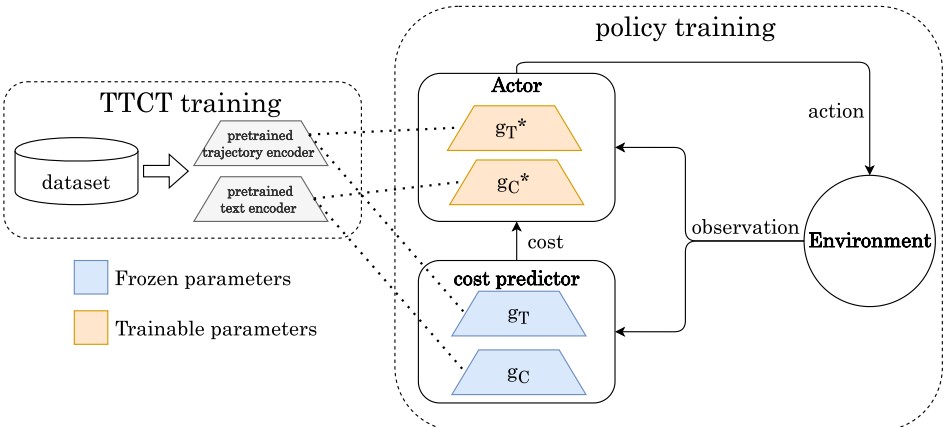

Figure 8: A diagram illustrating the usage of $g_T$, $g_C$ and $g_T^*$, $g_C^*$ when training policy.

```
# state_encoder                - CNN or DNN
# action_encoder               - DNN
# trajectory_encoder           - Causal Transformer
# text_encoder                 - Any Text Transformer
# episodic_cost_predictor      - DNN
# single_step_cost_predictor   - DNN
# labels_MC                    - [n , n] labels for multimodal
# W_tra[d_tra, d_e]            - learned proj of trajectory to embed
# W_t[d_t, d_e]                - learned proj of text to embed
# S[n, l, d_s]                 - minibatch of n state sequences
# A[n, l, d_a]                 - minibatch of n action sequences
# T[n, w]                      - minibatch of n text constrains

# extract feature representations of different inputs
S_f = state_encoder(S)
A_f = action_encoder(A)
Tra_f = trajectory_encoder(Concat([S_f, A_f], dim=-1))      # [n, l, d_tra]
T_f = text_encoder(T)                                       # [n, 1, d_t]

# normalization
T_e = l2_normalize(np.dot(T_f, W_t), axis=-1)               # [n, 1, d_e]
Tra_e = l2_normalize(np.dot(Tra_f, W_tra), axis=-1)         # [n, l, d_e]

# cost assignment
EpCost = episodic_cost_predictor(T_e.detach())              # [n]
logits_WT = torch.matmul(Tra_e * np.exp(t), T_e)            # [n, l, 1]
atten_score = sigmoid(logits_WT)                           # [n, l, 1]
Tra_atten = atten_score * Tra_e
cost = single_step_cost_predictor(Concat([Tra_atten[:, :l-1, :], T_e.repeat(1, l-1, 1).detach()],
                                                                      dim=-1)) # [n, l]

sum_cost = sum(cost, dim=1) # [n]
loss_CA = (mse_loss(EpCost, sum_cost) + mse_loss(sum_cost, EpCost))/2

# within-trajectory (WT)
labels_WT = torch.tensor([l-1] * n) # only the last time step is positive
loss_WT = cross_entropy_loss(logits_WT, labels_WT)

# multimodal contrastive (MC)
logits_MC = np.dot(Tra_e[:, -1, :] * np.exp(t), T_e) # [n, n]
loss_tra = KL_divergence_loss(logits_MC, labels_MC, axis=0)
loss_t = KL_divergence_loss(logits_MC, labels_MC, axis=1)
loss_MC = (loss_tra + loss_t)/2

# total loss
loss = loss_MC + loss_WT + loss_CA
```

Figure 9: Pseudocode for the code of training our TTCT.

# B Additional Experiments

## B.1 Violations Prediction Capability of Text-Trajectory Alignment Component

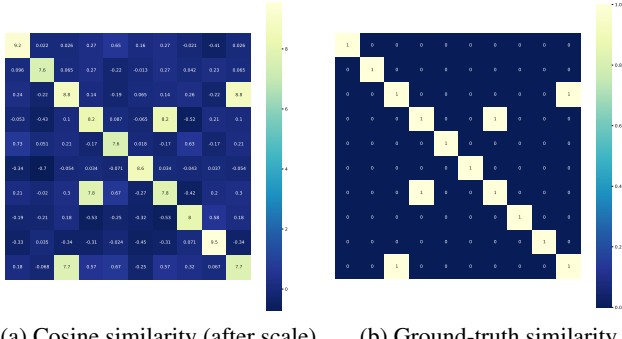

(a) Cosine similarity (after scale)    (b) Ground-truth similarity

Figure 10: Heatmap of cosine similarity between trajectory and text embeddings.

To further study the ability of our text-trajectory alignment component to predict violations, we conduct an experiment given a batch of trajectory-text pairs and we use the text-trajectory alignment component to encode the trajectory and textual constraint, and then calculate the cosine distance with Equation 5 between every two embeddings across two modal. We plot a sample of heatmap of calculated cosine similarity and ground-truth as presented in 10. Further, We plot the receiver operating characteristic (ROC) [8] curve to evaluate the performance of the text-trajectory alignment component as presented in Figure 11. AUC (Area Under the Curve) [8] values indicate the area under the ROC curve. The AUC value of our violations prediction result is 0.98. Then We set threshold $\beta$ equal to the best cutoff value of the ROC curve. We determine whether the trajectory violates a given textual constraint by:

$$\begin{cases} yes, & \text{if } \text{sim}(\tau, y) \geq \beta, \\ no, & \text{otherwise} \end{cases}, \tag{14}$$

The results are shown in Table 4. These results indicate that our text-trajectory alignment component can accurately predict whether a given trajectory violates a textual constraint.

Table 4: Violations prediction results of text-trajectory alignment component.

| Accuracy | Recall | Precision | F1-score |
|----------|--------|-----------|----------|
| 0.98 | 0.9824 | 0.8079 | 0.8866 |

## B.2 Case Study of Cost Assignment (CA) Component

We visualize the assigned cost for every state-action pair to demonstrate that the cost assignment component could capture the subtle relation between the state-action pair and textual constraint. Our intuition is that we should assign larger costs to state-action pairs that lead to violations of trajectory-level textual constraints, and smaller or negative values to pairs that do not contribute to constraint violations. Using the Hazard-World-Grid environment as an example, we choose three different types of constraints to show our results in Figure 12. The first row shows the textual constraint, the second row shows the trajectory of the agent in the environment, and to make it easier to visualize, we simplify the observation $s_t$ by representing it as a square, denoting the entity stepped on by the agent at time step $t$. The square emphasized by the red line indicates the final entity that makes the agent violate textual constraint at time step $T$. The third row shows the predicted cost of the agent at every time step $t$ and deeper colors indicate larger cost values.

The (a) constraint is **mathematical textual constraint:** *"You only have 20 HP. Lava and grass are dangerous, they will make you lose 3 and 2 HP, respectively. However, water can regenerate 1 HP. Please don't die.".* This constraint describes the two dangerous entities lava and grass, and the beneficial entity water. From the third-row heat map, we can observe that our cost assignment

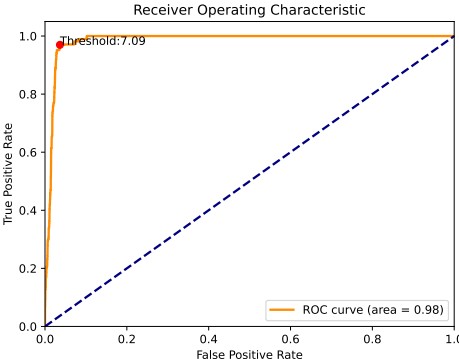

Figure 11: **ROC curve of text-trajectory alignment component.** The x-axis represents the false positive rate, and the y-axis represents the true positive rate. The closer the AUC value is to 1, the better the performance of the model; conversely, the closer the AUC value is to 0, the worse the performance of the model.

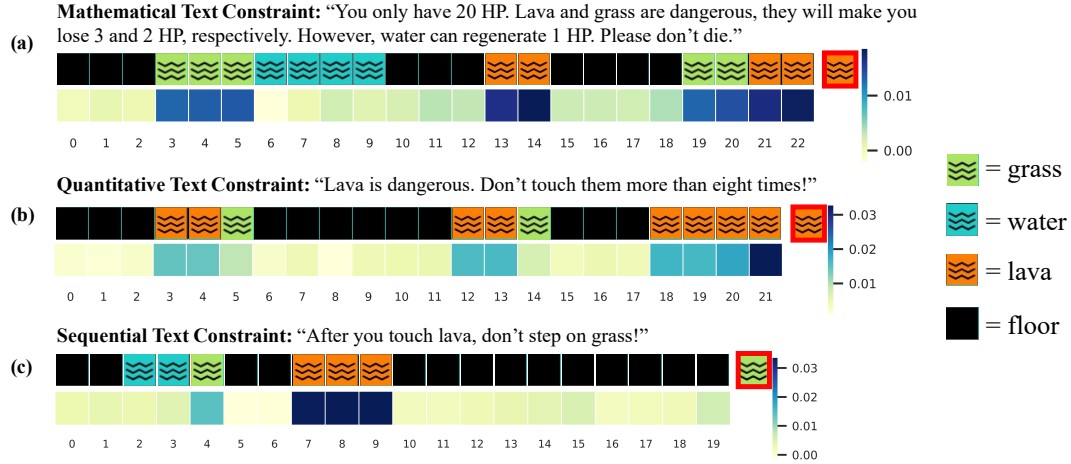

Figure 12: **Case study of cost assignment component on three types of textual constraints.** The first row of every case shows the textual constraint, the second row shows the trajectory of the agent in the environment and each square represents the object stepped on by the agent at that time step, the third row shows the assigned cost of the agent at each time step, and the fourth row shows the time steps. The red line indicates the final observation where the agent violates the textual constraint.

component assigns a high cost to the action that steps on lava or grass, with the cost increasing as the agent approaches the constraint-violating situation. Not only that, the CA component also recognizes the different levels of danger posed by lava and grass. Since stepping on lava will deduct 3 HP while stepping on grass will deduct 2 HP, the CA component assigns a larger cost value at time step $13 - 14$ compared to the cost value at time step $3 - 5$.

The (b) constraint is **quantitative textual constraint**: *"Lava is dangerous. Don't touch them more than eight times!"*. When stepping on the floor, the CA component considers these actions to be safe and assigns a cost of nearly 0. However, when the agent steps onto lava, it assigns a higher cost, especially when the agent steps on lava for the eighth time. At this point, our CA component concludes that the situation has become extremely dangerous, and one more step on lava will violate the constraint, thus giving the highest cost compared to the before time steps.

The (c) constraint is **sequential textual constraint**: *"After you touch lava, don't step on grass!"*. Our CA component captures two relevant entities: lava and grass, and understands the sequential relationship between entities. When the agent first steps onto the grass, the text-trajectory alignment component determines that this action does not violate the textual constraint. However, after the

agent steps onto lava and then steps onto the grass for the second time, the text-trajectory alignment component detects the cosine similarity $\text{sim}(\tau, y)$ greater than the threshold $\beta$, thereby violating the textual constraint. And CA component captures that the key trigger condition for violating constraint is stepping onto the lava, therefore assigning a relatively larger cost to such actions at time step $7 - 9$. The cost assignment component also assigns relatively small costs to some safe actions that are stepping on the floor, such as steps $13 - 15$. This is because it detects that the agent, although hasn't stepped on grass yet, is trending towards approaching grass, which is a hazardous trend, thus providing a series of gradually increasing and small costs. This demonstrates that our component not only monitors key actions that lead to constraint violations but also monitors the hazardous trend and nudging the agent to choose relatively safer paths.

### B.3  Results for Different Types of Constraints

To evaluate the agent's understanding of different types of trajectory-level textual constraints, we conducted an additional experiment using the CPPO_PID algorithm. During training, we separately tracked the average episodic reward and the average episodic cost for three types of textual constraints as presented in Figure 13. From the learning curves, we can observe that for every type of constraint, our CP mode can achieve the lowest violation rate compared to CP without the CA component mode and ground-truth cost (GC) mode.

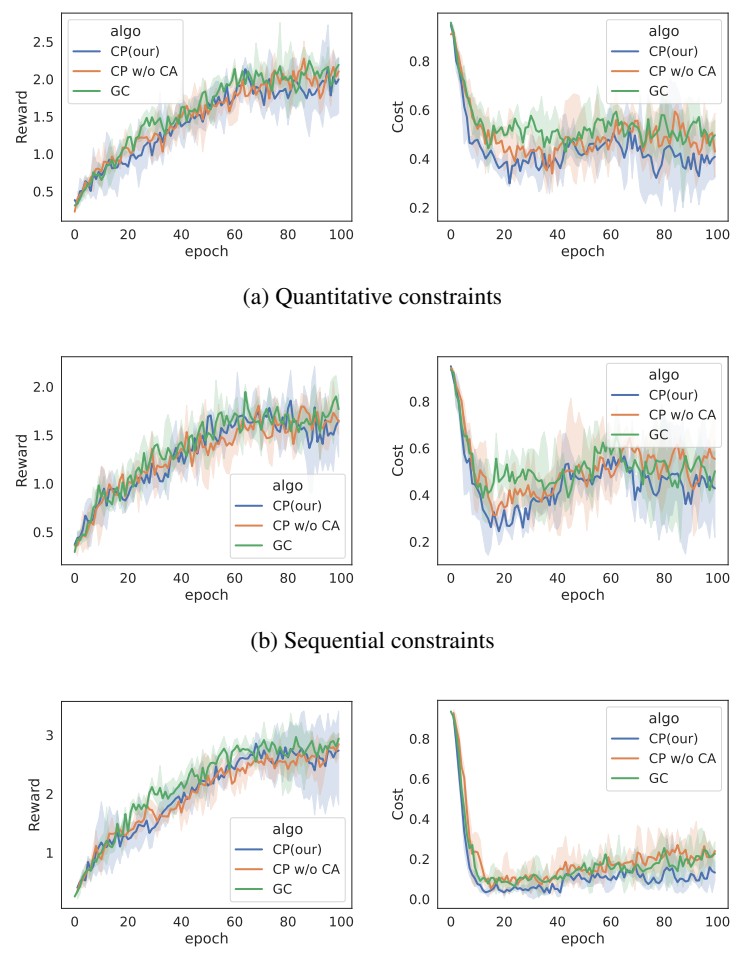

Figure 13: Learning curves for different types of textual constraints. The left figure shows the average reward and the right figure shows the average cost. The result shows for each type of textual constraint, the policies trained by predicted cost from the TTCT can achieve lower constraint violations.

## B.4 Inference time

We perform the trajectory length sensitivity analysis on Hazard-World-Grid. Since our framework is mainly used for reinforcement learning policy training where data is typically provided as input in batches, we counted the inference time for different trajectory lengths with 64 as the batch size, using the hardware device V100-32G. Figure 14 shows that the average inference time per trajectory is 10ms for trajectories of length 100.

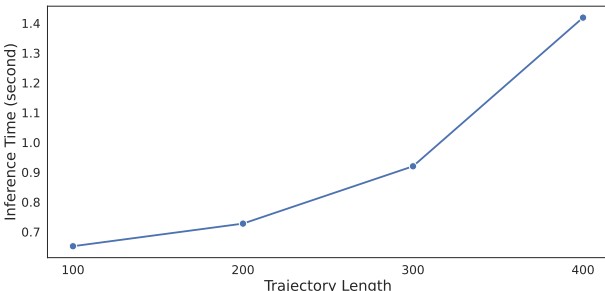

Figure 14: Inference time of different trajectory lengths for Hazard-World-Grid on the V100-32G hardware device. Batch size is 64.

## B.5 Different Text Encoder

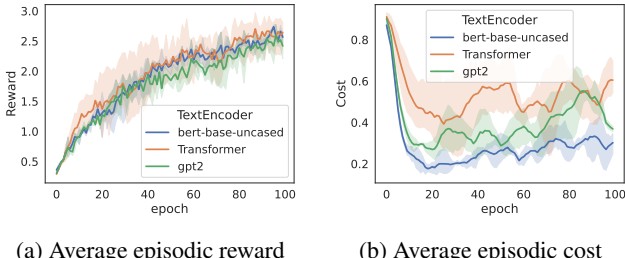

(a) Average episodic reward     (b) Average episodic cost

Figure 15: Empirical analyses on Hazard-World-Grid with varying text encoders. We choose three different models, transformer-25M [41], gpt2-137M [31], and bert-base-uncased-110M [7].

## B.6 Additional Results

We present the learning curve of our main experiment in Figure 4. We also present the main results and ablation study in Table 5 and 6.

Table 5: **Evaluation results of our proposed method TTCT.** ↑ means the higher the reward, the better the performance. ↓ means the lower the cost, the better the performance. Each value is reported as mean ± standard deviation and we shade the safest agents with the lowest averaged cost violation values for every algorithm.

| Task | Stats | PPO_Lag | | CPPO_PID | | FOCOPS | | PPO |
|------|-------|---------|---|----------|---|--------|---|-----|
| | | CP(our) | GC | CP(our) | GC | CP(our) | GC | |
| Grid | Avg.R ↑ | $2.70_{\pm0.11}$ | $2.71_{\pm0.11}$ | $2.01_{\pm0.14}$ | $2.00_{\pm0.26}$ | $1.52_{\pm0.12}$ | $1.53_{\pm0.28}$ | $2.55_{\pm0.23}$ |
| | Avg.C ↓ | $0.28_{\pm0.08}$ | $0.61_{\pm0.07}$ | $0.28_{\pm0.03}$ | $0.40_{\pm0.07}$ | $0.48_{\pm0.20}$ | $0.58_{\pm0.12}$ | $0.88_{\pm0.05}$ |
| Goal | Avg.R ↑ | $0.23_{\pm0.12}$ | $0.22_{\pm0.29}$ | $0.36_{\pm0.20}$ | $0.35_{\pm0.77}$ | $0.02_{\pm0.36}$ | $-0.63_{\pm0.68}$ | $0.23_{\pm0.55}$ |
| | Avg.C ↓ | $0.19_{\pm0.02}$ | $0.26_{\pm0.03}$ | $0.11_{\pm0.03}$ | $0.35_{\pm0.20}$ | $0.10_{\pm0.03}$ | $0.41_{\pm0.07}$ | $0.32_{\pm0.19}$ |

Table 6: **Ablation study of removing the cost assignment (CA) component.** ↑ means the higher the reward, the better the performance. ↓ means the lower the cost, the better the performance. Each value is reported as mean ± standard deviation and we shade the safest agents with the lowest averaged cost violation values for every algorithm.

| Task | Stats | PPO_Lag | | CPPO_PID | | FOCOPS | | PPO |
|------|-------|---------|---|----------|---|--------|---|-----|
| | | CP(Full) | CP w/o CA | CP(Full) | CP w/o CA | CP(Full) | CP w/o CA | |
| Grid | Avg.R ↑ | $2.70_{\pm0.11}$ | $2.57_{\pm0.11}$ | $2.01_{\pm0.14}$ | $2.00_{\pm0.16}$ | $1.52_{\pm0.12}$ | $0.79_{\pm0.06}$ | $2.55_{\pm0.23}$ |
| | Avg.C ↓ | $0.28_{\pm0.08}$ | $0.56_{\pm0.08}$ | $0.28_{\pm0.03}$ | $0.52_{\pm0.14}$ | $0.48_{\pm0.20}$ | $0.43_{\pm0.04}$ | $0.88_{\pm0.05}$ |
| Goal | Avg.R ↑ | $0.23_{\pm0.12}$ | $-0.62_{\pm0.10}$ | $0.36_{\pm0.20}$ | $-0.08_{\pm0.97}$ | $0.02_{\pm0.36}$ | $-0.63_{\pm0.21}$ | $0.23_{\pm0.55}$ |
| | Avg.C ↓ | $0.19_{\pm0.02}$ | $0.35_{\pm0.10}$ | $0.11_{\pm0.03}$ | $0.29_{\pm0.04}$ | $0.10_{\pm0.03}$ | $0.23_{\pm0.09}$ | $0.32_{\pm0.19}$ |

# C  Broader Impacts and Limitation

Our method can help train agents in reinforcement learning tasks with total free-form natural language constraints, which can be useful in various real-world applications such as autonomous driving, robotics, and game playing. There are still limitations to our work. Our method may not be able to completely eliminate constraint violations. Our method has the contextual bottleneck as the length of the trajectory increases. We performed a trajectory length sensitivity analysis on Hazard-World-Grid. As shown in Figure 16, initially increasing the trajectory length improves performance because longer trajectories may provide more dependencies. However, beyond a certain point, further increases in trajectory length result in a slight drop in AUC. This decline is because the trajectory encoder has difficulty capturing global information. We consider this bottleneck to be related to the transformer's encoding capability.

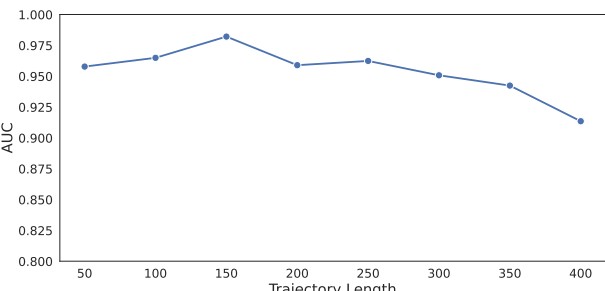

Figure 16: Evaluation results of different trajectory lengths for Hazard-World-Grid. Sets of trajectories with varying lengths shared the same set of textual constraints.

