# OpenReview forum: "From Text to Trajectory: Exploring Complex Constraint Representation and Decomposition in Safe Reinforcement Learning"
_NeurIPS.cc/2024/Conference — NeurIPS 2024 poster_

### Official Review · Reviewer_T5E9 · 2024-07-10

**Soundness:** 3
**Presentation:** 3
**Contribution:** 4
**Rating:** 7
**Confidence:** 4

**Summary:**

This paper introduces a type of universal natural language constraint to model diverse real-world safety requirements. Also, to avoid using the specific human-designed cost function, this paper introduces a Unified Trajectory-Level Textual Constraints Translator (U3T) for aligning text with trajectories and assigning costs via attention mechanisms. By conducting experiments on two environments, this paper reaches the conclusion that U3T can accurately predict whether a given trajectory violates the natural language constraint, and the policy trained with predicted cost can behave more safely.

**Strengths:**

1. Broad applicability: U3T introduces trajectory-level textual constraints, capable of modeling diverse constraint requirements in real-world scenarios, making it more widely applicable across different types of constraints and complex environments compared to traditional methods.
2. Novelty: Addressing the natural language constrained reinforcement learning challenge through a novel approach involving multimodal alignment and novelly integrating a credit assignment component within the framework.
3. Automated constraint handling: Through its text-trajectory alignment and attention-based cost assignment components, U3T automates constraint handling and cost allocation, reducing the need for manual design of cost functions and enhancing system flexibility and generality.
4. Empirical results demonstrate that policies trained with U3T achieve lower violation rates compared to standard cost functions.

**Weaknesses:**

1. Complexity and computational cost: Implementing U3T involves complex multimodal learning and attention mechanisms, which may require significant computational resources and time costs, especially when dealing with large-scale datasets or real-time applications.
2. The authors do not specify exactly what model of text encoder was used in the experiments.
3. Data Dependency: Effective implementation of U3T heavily relies on high-quality and well-annotated textual and trajectory data, which may be challenging to acquire and curate in some practical applications.

**Questions:**

1. Is it feasible for the authors to experimentally analyze the inference speed of the proposed framework?
2. Could the authors specify the model of the text encoder used in the experiments and provide an in-depth exploration of its selection rationale?
3. Leveraging large language models for text understanding shows promise. Have the authors considered strategies for integrating these models to augment the comprehension and deconstruction of intricate textual constraints?

**Limitations:**

The authors have discussed the limitations in the paper.

---

> ### Author Rebuttal · Authors · 2024-08-07
>
> Thanks for the constructive comments. We are grateful to the the reviewer's conductive comments. We will answer the questions below.  We believe we addressed all the concerns and are glad to follow up in the discussion phase.
>
> ----
>
> **Q1: Inference speed**
>
> **Answer:** We perform the trajectory length sensitivity analysis on Hazard-World-Grid. Since our framework is mainly used for reinforcement learning policy training where data is typically provided as input in batches, we counted the inference time for different trajectory lengths with 64 as the batch size, using the hardware device V100-32G. Figure 3 in the rebuttal PDF shows that the average inference time per trajectory is 10ms for trajectories of length 100. This inference time is generally acceptable.
>
> ---
>
> **Q2: The text encoder we used**
>
> **Answer:** Due to its bidirectional encoding capabilities, Bert [1] can better understand the context of words, and in particular, excels in capturing dependencies within sentences. While autoregressive language models such as GPT2 are more suitable for text-generation tasks [2]. So we choose bert-base-uncased as the text encoder. Sorry for omitting the description of text encoder architecture in our paper.  We will add it in the final version.
>
> [1] Devlin, J., Chang, M. W., Lee, K., & Toutanova, K. (2018). Bert: Pre-training of deep bidirectional transformers for language understanding. *arxiv preprint arxiv:1810.04805*.
>
> [2] Ethayarajh, K. (2019). How contextual are contextualized word representations? Comparing the geometry of BERT, ELMo, and GPT-2 embeddings. *arXiv preprint arXiv:1909.00512*.
>
> ----
>
> **Q3: Leverage Large language model**
>
> **Answer:** Thanks to the reviewer for the constructive view.  Large language models exhibit powerful semantic understanding and task decomposition capabilities, so in the future, LLM could be utilized to decompose more complex constraints into several simple constraints, which can then be handed off to lower-level constraint-obey agents for execution.
>
> ----
>
> *We hope these responses address your concerns. We remain open to further discussion.*

---

> > ### Comment · Reviewer_T5E9 · 2024-08-12
> > **Official Comment by Reviewer T5E9**
> >
> > Thanks for the authors' response. I have no further concern and will lean to keep the score.

---

### Official Review · Reviewer_z2Bi · 2024-07-18

**Soundness:** 3
**Presentation:** 3
**Contribution:** 2
**Rating:** 6
**Confidence:** 3

**Summary:**

* The paper proposes U3T, a new system for more robust safe RL under general text constraints.
* The key innovation of the paper is to generalize text constraints such that the constraints don’t refer to a specific entity/state and addressing cost sparsity where constraints are only violated at the final time step.
* The proposed system consists of a trajectory encoder and text encoder that are trained jointly using a contrastive loss defined by the KL-divergence.
* The authors solve cost sparsity by identifying actions/states that contribute to the violation of a constraint in an attention mechanism.

**Strengths:**

* The paper solves an interesting and intuitive problem. There is a clear application of generalizing text constraints from single entity to general entities.
* The formulation is intuitive and well-explained. The components introduced build on a simple and effective design
* Ablation studies seem to demonstrate the benefit of the cost assignment portion of their system design, a portion that could be extremely expensive cost-wise.
* The general cost (Figure 2/Figure 3) of their method seems to reduce in comparison to their baselines that use ground truth.

**Weaknesses:**

* I found some details missing in the text. I discuss some portions that could use clarification here.
    * Which text encoder was used in this experiment? I couldn’t find this in the paper and from your appendix section A.4, it seems untrained. You refer to finetuning with LORA though so I was confused. A small nit, could be interesting to connect the text encoder performance to results. I realize space is tight but if these could be incorporated into the main paper, that would clarify and make the experiments easier to interpret.
    * I’m unsure how to interpret the average reward results in comparison to the cost results. See questions. My intuition was that rewards should be larger given the pareto frontier results.
* I’m having trouble differentiating this paper from the Lou et. al. paper. I took a quick look at Lou et. al. --  I can understand that entities needed to be modeled in the paper but from what I can tell, that paper could also be applied in the settings you considered given the use of the pretrained LLM. From Figure 1, I couldn’t see the specific entity problem you were mentioning in the paper. Why wasn’t this added as a baseline? Could you provide more discussion on why these two are different? I think the approach in this paper has considerable novelty so I won’t say that this is entirely negative but it does diminish the contribution slightly.

**Questions:**

* I noticed that generally, while the cost of the method would decrease, the reward wouldn’t change significantly (Figure 2). I actually found this a bit unintuitive. I figured that lower cost would imply a longer trajectory and therefore, longer windows to collect reward over, so average reward could increase. I found this discussion missing in the text. Could this be elaborated?
* How long were the trajectories used in the paper? (What was the average size of T?) Could there be a context bottleneck?

**Limitations:**

* See question about context bottleneck. Could this be a limitation to discuss?

---

> ### Author Rebuttal · Authors · 2024-08-07
>
> Thank you for your thoughtful review. And we are very pleased that you appreciate the effectiveness and direction of our research.  We would like to address your concerns below.
>
> ---
>
> **W1.1: About text encoder**
>
> **Answer:** We used pre-trained bert-base-uncased [1] as a text encoder. During U3T training, we fully fine-tuned the text encoder. The text encoder was fine-tuned again using LoRa during RL policy training. The original U3T, used for predicting costs, was not retrained during RL policy training. A formal description is provided in Algorithm 1.
>
> We have added related experiments. We chose three different models: untrained transformer-25M, pretrained gpt2-137M, and pretrained bert-base-uncased-110M.
>
> For natural language understanding tasks, bert-base-uncased has the strongest capability, GPT-2 comes second, and Transformer is the weakest [2]. As shown in Figure 3 of the rebuttal PDF, using bert-base-uncased as a text encoder yields the best results, indicating that models with stronger semantic understanding capabilities improve our framework's performance.
>
> [1] Devlin, J., Chang, M. W., Lee, K. et. al. Bert: Pre-training of deep bidirectional transformers for language understanding.
>
> [2] Ethayarajh, K. (2019). How contextual are contextualized word representations? Comparing the geometry of BERT, ELMo, and GPT-2 embeddings.
>
> ----
>
> **W1.2&Q1: The reward didn't change significantly.**
>
> **Answer:** Safe RL problems can be classified into two categories (more formally defined in [1]).
>
> The first category involves cases where maximizing reward and minimizing cost are aligned. For example, in an environment with a ball in the center of a 3D square, the robot must touch the ball without stepping outside the square (Dalal et al. [2]). Here, touching the ball (maximizing reward) aligns with staying inside the square (minimizing cost). This type of problem is simpler, as it reduces to a standard RL problem focused on maximizing reward. The second category involves scenarios where maximizing reward and minimizing cost are not aligned. In these cases, the policy that maximizes rewards is not the one that minimizes costs, leading to a trade-off between reward and cost. This type of problem is more challenging, as avoiding hazards does not provide an intuitive reward gain.
>
> Our experimental benchmark falls into the second category. In our environment, hazards often lie between the agent and the target. To reach the goal without violating constraints, agents must take detours and adopt behaviors that ensure safety but do not directly yield rewards. This explains why a low violation rate and extended reward collection time do not significantly boost rewards—the additional time is spent avoiding hazards.
>
> In our setup, minimizing costs does not directly imply maximizing rewards. Achieving a lower violation rate with the same rewards is already an improvement. This challenge is also related to the inherent limitations in current safe RL algorithms, which often lack mechanisms for enhancing exploration while ensuring safety. We appreciate the reviewer's unique perspective and will discuss this in detail in the final version.
>
> [1] Liu, Z., Guo, Z., Yao, Y., Cen, Z., Yu, W., Zhang, T., & Zhao, D. Constrained decision transformer for offline safe reinforcement learning.
>
> [2] Dalal, G., Dvijotham, K., Vecerik, M., Hester, T. et. al. Safe exploration in continuous action spaces.
>
> ----
>
> **W2: About Lou et. al. paper**
>
> **Answer:** Thank you to the reviewer for recognizing the novelty of our paper. We believe it is the first to enable the modeling of fully generalized textual constraints. Our innovation lies in our unified framework that handles constraints with complex logic involving multiple states, entities, and timesteps. Lou et al.'s work determine constraint violations by calculating the similarity of **single** observations to constraints, limiting them to single-state or single-timestep constraints. This approach cannot model complex real-world constraints, such as "don't drive after drinking," which involves sequence and dependency.
>
> Our work, in contrast, addresses global trajectories, allowing us to model a broader range of constraints. We focus on universal textual constraints with complex semantics, which Lou et al.'s approach cannot handle due to their lack of components to model trajectory dependencies and align trajectory semantics with textual semantics. Hence, we did not use their work as a baseline for our research.
>
> We conducted comparative experiments in the hazard-world-grid environment to verify our method's accuracy under their task settings. We applied constraints to single states and accumulated costs for violations instead of terminating the episode. Since their training code is not open-sourced, we obtained their experimental results from their paper's figures. The experimental results are as follows:
>
> | Method      | Avg. Reward     | Avg. Cost      |
> | ----------- | --------------- | -------------- |
> | Lou et. al. | $4.9_{\pm0.2}$  | $4.5_{\pm0.5}$ |
> | our         | $4.92_{\pm0.3}$ | $3.3_{\pm0.4}$ |
>
> ----
>
> **Q2: About Trajectory length**
>
> **Answer:** The maximum trajectory length in our experiment is 200, with an average episode length of approximately 100, depending on the violation rate. We conducted a trajectory length sensitivity analysis as shown in Figure 2 on Hazard-World-Grid, using AUC to measure prediction accuracy. Initially, increasing the trajectory length improves performance as longer trajectories provide more dependencies. However, beyond a certain point, further increases result in a slight drop in AUC due to the transformer's difficulty in capturing global information, indicating a contextual bottleneck related to the transformer's encoding capability. We will add the description of trajectory length and discuss the context bottleneck in the final version.
>
> ----
>
> *We hope these responses address your concerns. We remain open to further discussion.*

---

> > ### Author Response · Authors · 2024-08-12
> >
> > Dear Reviewers z2Bi,
> >
> > We want to make sure that our reviews address your concerns. If you need further clarification, please feel free to contact us. Thank you for taking the time to review our papers.
> >
> > Sincerely,
> >
> > Authors.

---

### Official Review · Reviewer_cY35 · 2024-07-20

**Soundness:** 1
**Presentation:** 2
**Contribution:** 2
**Rating:** 5
**Confidence:** 3

**Summary:**

- This work proposes an approach to training RL agents with constraints. The proposed approach learns language embeddings of constraints and embeddings of trajectories. During training, a similarity score is used to align the space of constraint embeddings and the space of trajectory embeddings.
- Moreover, the language embedding of the constraint is also used to do fine-grained credit assignment across the actions in a trajectory. The authors find that this improves performance.

**Strengths:**

- The approach is novel in that the technique of credit assignment is applied to the constraint rather than the reward. Or at least this is the claim, and I was not able to find otherwise.
- The approach is shown to be effective. With an ablation study, the authors show that their credit assignment approach gives better performance
- I have questions about the setup and novelty (see weaknesses), but overall this work seems to represent a positive contribution: the authors show that it is possible to encode trajectory-level constraints with text and to train an RL agent to respect those constraints.

**Weaknesses:**

- I have a hard time understanding where this work stands in relation to prior works. The main contribution seems centered around the approach of encoding trajectory level constraints using natural language. But both trajectory level constraints and natural language constraints have been explored independently. And the combination of these two aspects seems straightforward: encode the trajectory level constraint using natural language. I may be overlooking something that makes this less straightforward. I don't quite follow the explanation for novelty on lines 101-104. The Related Works section mentions prior work which learns language representations of constraints, but says that these works do not consider trajectory-level constraints. But because the constraints are given in natural language, it doesn't seem that anything in principle prevents these prior approaches from doing so? What is the technical innovation of the proposed work that allows for trajectory level constraints?
- A few things seem missing from the setup (see questions)
- I have concerns about the soundness of the described approach. Where does the ground truth come from when training the text-trajectory alignment component? As far as I can tell, there is notion of a textual constraint but not a notion of a ground truth verification of a constraint. From what I understand of appendix 1, the textual constraint is determined to be violated by a discriminator model (see questions). In any case, the authors should explain how constraints are determined to be violated. This matters because it means that the discriminator model and the agent could both make the same systematic linguistic errors.
- I have concerns about the soundness of the approach in general. Using natural language to describe constraints seems at odds with the enterprise of making RL more "safe." Natural language can be ambiguous, and machine learned representations of language can be imperfect, and prone to making semantic errors. In contrast to other approaches which provide formal guarantees on behavior [1], this seems much less safe. However, I am conscious that others in the field think differently, given the prior work.

## minor
- Figure 3: Labels are missing in the legend
- Line 297: "Table" --> "Figure"

## references
[1] Fulton, Nathan, and André Platzer. "Safe reinforcement learning via formal methods: Toward safe control through proof and learning." Proceedings of the AAAI Conference on Artificial Intelligence. Vol. 32. No. 1. 2018.

**Questions:**

- Line 174: What is a "negative pair"? Is it a constraint and a trajectory that violates that constraint?
- Line 175: Given that the mapping from trajectories to constraints is many-to-many, doesn't $q$ need to be normalized so that the probabilities sum to 1? That is, there are many positive pairs that contain the same constraint, so they should all receive an equal share of positive probability.
- Fig 5: I think I'm missing something basic: how are is the Pareto front determined? I could assume that per cost, the best reward possible can be determined by some procedure? Is this what was done?
- Section 6: Where do the textual constraints come from? Are they human annotated? Appendix 1 seems to suggest not.
  - Is it guaranteed that the discriminator can tell when a textual constraint is violated?
- Line 289: What is the difference between ground-truth mode and cost prediction mode? Is it the cost assignment?

**Limitations:**

Limitations are mentioned in the appendix.

---

> ### Author Rebuttal · Authors · 2024-08-07
>
> We thank the reviewer for carefully reviewing our submission. We address each point of your concerns separately below.
>
> -----
>
> **W1: Relation to prior works**
>
> **Answer:** We appreciate the reviewer’s constructive question. See the overall response about the **relationship to previous work**. We will elaborate on the above discussion in the final version.
>
> ----
>
> **Q1: About negative pair**
>
> **Answer:** In our formal setting, "negative pair" means an unmatched pair of constraint and trajectory, that is no violation of this textual constraint in the trajectory.
>
> ----
>
> **Q2: Probability normalization**
>
> **Answer**: Sorry for causing the misunderstanding. We do normalization in our experiments:
>
> ```label_probs = F.softmax(label, dim = 1)```
>
> We will rewrite the loss function part for clarity in the final version
>
> ----
>
> **Q3: About Pareto front**
>
> **Answer:** During RL training, we collected about 200 policies and evaluated them on 50 new episodes for average cost and reward scores. We follow the definition in [1]. To determine the Pareto front, we compared each policy with others. A policy is not on the Pareto front if another policy has equal or better rewards and equal or lower costs. If no such policy exists, the current policy is part of the Pareto front.
>
> So for a given cost, the policy with the highest reward may not be on the Pareto front if another policy exists that outperforms it in both reward and cost.
>
> [1] Wikipedia contributors. (2024, June 12). Pareto front. In *Wikipedia, The Free Encyclopedia*.
>
> ----
>
> **Q4.1: Where do the textual constraints come from**
>
> **Answer:** To ensure the generation of textual constraints is both professional and effective, we implemented the following steps:
>
> 1. We developed a descriptor function to systematically identify all hazards encountered at every timestep when random policy explores the environment. We identified the hazards by checking the system state and storing them in a structured text format. For example:
>
>    ```
>    hazards = {
>        "timestep":{
>            5: "step into chemical area",
>            # at 5th timestep, the agent step into chemical area
>            12: "step on the grass",
>            20: "step into the water",
>            30: "step into the lava"
>        }
>    }
>    ```
>
> 2. We then constructed various logical combinations of different hazards. And combination format depends on the type of constraint we want to generate. Here we take the mathematical constraint as an example, since it was checked that the agent touched four hazardous items, we flexibly assigned a lost HP value to each hazard and used the total amount of HP lost in the trajectory, as the total HP:
>    ```
>    combination = {
>        "constraint type": "Mathematical",
>        "HP lose":{
>            "chemical area": 3,
>            # step into chemical area will lose 3 HP
>            "lava": 2,
>            "grass": 1,
>            "water": 0
>        }
>        "total HP": 6
>    }
>    ```
>
> 3. We engaged several researchers to define a large number of constraint templates in diverse language styles, which were used to rephrase the logical combinations into unstructured natural language.
>
> These steps enabled us to generate a series of textual constraints, each corresponding to a structured combination. In practical applications, generating textual constraints could be simplified as follows:
>
> - Instance 1: Researchers directly describe constraints based on video/image demonstrations of a trajectory.
> - Instance 2: Gather videos with dangerous scenes and textual commentary, convert these into trajectories and natural language constraints, and use U3T to train agents to adhere to real-life constraints.
>
> ----
>
> **W3&Q4.2: Soudness of discriminators**
>
> **Answer:** Sorry for the misunderstanding due to the omission of the detailed description of the discriminator. We manually designed (not trained) complex discriminators for various types of constraints which are absolutely accurate. Each natural language constraint corresponds to a structured logical combination (mentioned in the Q4.1). Human-designed discriminators assess whether a trajectory violates the logical combination, ensuring accuracy and the soundness of our approach. We will provide further details in the final version. It's important to note that this complex discriminator is not needed for the practical application of U3T; it was designed solely to evaluate U3T's performance in our study.
>
> ----
>
> **Q5: Difference between ground-truth mode and cost prediction mode**
>
> **Answer:** As we mentioned in Section 6.1 of the paper, in the ground-truth mode, policies are trained with human-designed constraint discriminators, which will give accurate cost at the timestep when the constraint is violated. In the cost prediction mode, there are two types of cost, the constraint violation cost and the assigned cost, they are all predicted by U3T. The predicted cost function is given by Equation 13 in the paper.
>
> ----
>
> **W4: The soundness in general**
>
> **Answer:** The reviewer raises an interesting point.
>
> Our work emphasizes the flexibility and generalization of language. We propose a modeling paradigm for safe reinforcement learning (RL) that integrates future advancements in natural language and series modeling, enhancing RL safety over time. Rapid adaptation and flexibility can be crucial in dynamic environments, where natural language constraints may be more effective than predefined formal ones.
>
> We align agents' behavior with human intentions similar to OpenAI's RLHF [1]. Just as defining alignment safety in large language models is challenging, our method also struggles with providing a clear definition of textual constraint safety boundaries. So we focus on providing a prototype for safe RL with textual constraints and aim to develop a more rigorous definition of textual constraint safety in future work.
>
> [1] OpenAI. ChatGPT: Optimizing Language Models for Dialogue, 2022.
>
> ----
>
> *We hope these responses address your concerns.*

---

> > ### Author Response · Authors · 2024-08-12
> >
> > Dear reviewer cY35,
> >
> > We hope this message finds you well. We want to check if our reviews address your concerns. If you have any further questions or if there’s anything you’d like to discuss, please feel free to comment further. Thank you for your time again.
> >
> > Sincerely,
> >
> > Authors.

---

> > > ### Comment · Reviewer_cY35 · 2024-08-13
> > > **Response to authors**
> > >
> > > I thank the authors for their response. I will keep my score. I think I understand the argument for novelty that the authors are making a little better. But there are still some gaps for me. The rebuttal says that "previous methods fail to offer a unified framework for trajectory-level textual constraints," but I don't really understand what "unified framework" was lacking in the prior work. As long as a constraint can be specified in natural language, can it not be considered part of a "unified framework"?

---

> > > > ### Author Response · Authors · 2024-08-13
> > > >
> > > > **Thank the reviewer for raising this question. Regarding the concept of a "unified framework," we would like to clarify the following:**
> > > >
> > > > When we discuss a "unified framework," we are referring to the ability to consistently handle and represent textual constraints at the trajectory level. While previous methods could use natural language to describe constraints, they often required different state/trajectory encoding models or decision processes to handle different types of semantic constraints. This indicates that these methods lacked a unified framework capable of addressing complex logical associations in both spatial and temporal dimensions.
> > > >
> > > > **Specifically:**
> > > >
> > > > 1. **Limitations in the spatial dimension:** Previous methods mainly focused on state descriptions in the spatial dimension, meaning they could only check if a constraint was violated at specific states. This "point-based" approach fails to address sequential logic in the "line-based" temporal dimension, limiting the ability to describe and constrain long-term behavior within a trajectory.
> > > > 2. **Unification of the training method:** More importantly, our method no longer relies on manually labeled datasets for token classification within the text to determine which entity in the environment a constraint pertains to. This classification approach typically requires extensive manual labeling and is often restricted by specific semantic types. Our method leverages the text itself as a supervisory signal, which was not considered in previous work. The text not only describes constraints but also provides a unified supervisory training signal for trajectory encoding. This dual role allows our method to naturally handle various types of semantic constraints without needing specific model adjustments for different constraint types.
> > > >
> > > > Therefore, the so-called "unified framework" refers not only to the ability to express constraints in natural language but also to a system capable of uniformly encoding trajectories and handling various complex trajectory-level constraints, which our method achieves effectively.

---

### Author Rebuttal · Authors · 2024-08-07

We thank all reviewers for their time spent reviewing our paper and are grateful for the endorsement on **novelty** ("the approach is novel in that the technique of credit assignment is applied to the constraint" - cY35, "the paper solves an interesting and intuitive problem" - z2Bi, "through a novel approach" - T5E9) and **effectiveness** ("the approach is shown to be effective" - cY35, "effective design" - z2Bi, "achieve lower violation rates" - T5E9).

---

Here is our unified response to the common questions regarding the **relationship to previous work:**  in our paper, a trajectory-level constraint involves complex logic across multiple states, entities, and timesteps. Prior works focused on simpler constraints related to single states or timesteps, which limits their ability to model complex safety requirements. Despite constraints being provided in natural language, previous methods fail to offer a unified framework for trajectory-level textual constraints due to the lack of a unified representation of trajectory dependencies and the inability to align trajectory semantics with natural language.The novelty of our work lies in the unified understanding and application of the universal constraints. We align the factual logic in the global trajectory with the semantic logic in the text without requiring manual encoding or separate models for each type of constraint. We achieve this by utilizing **the supervision inherently present in the natural language**.

We provided individual answers to every reviewer. We will adapt the paper based on all reviewers' insightful comments and questions.  And we are happy to follow up in the discussion phase.

----

**Rebuttal pdf**

We provided additional experimental results to answer some of the reviewers' questions:

- Figure 1: question about text encoder from reviewer z2Bi
- Figure 2: question about trajectory length from reviewer z2Bi
- Figure 3: question about inference time from Reviewer T5E9

---

### Decision · Program_Chairs · 2024-09-25

**Decision:**

Accept (poster)

**Comment:**

Learning joint embeddings between text and trajectories is useful and topical.

Reviewers note that the main weakness is how the work relates to other research. Authors put this forward as a "unified framework" for reasoning about language and trajectories. With a focus on the fact that their method doesn't require manual annotations and alignment between the concepts in the trajectory and the linguistic concepts. But there is a tremendous literature that is being ignored which does exactly this.

I would really encourage the authors to engage with this literature because it points the way toward more interesting experiments for followup work. Reviewers point to work in safe RL, another area which does the same has to do with robots that follow linguist commands including those that incorporate notions of safety with LTL. This work often doesn't require annotations or alignment just like the work here.

In addition, there are so many other linguistic phenomena and logics to consider when aligning language to trajectories. Claiming that this is a unified framework for any constraints and notions of safety is far too much.

That being said, reviewers found that the topical nature of the work overrides these concerns and I agree. But it's to the author's benefit to resolve these issues with the manuscript. It will only allow them to engage with a much wider community.